# Polyphenols Profile and Antioxidant Activity Characterization of Commercial North Italian Ciders in Relation to Their Geographical Area of Production and Cidermaking Procedures

Federica Mainente [1,*], Simone Vincenzi [2], Corrado Rizzi [1] and Gabriella Pasini [2]

1 Department of Biotechnology, University of Verona, Strada Le Grazie 15, 37134 Verona, Italy; corrado.rizzi@univr.it
2 Department of Agronomy Food Natural Resources Animals and Environment, University of Padova, Viale dell'Università 16, 35020 Padova, Italy; simone.vincenzi@unipd.it (S.V.); gabriella.pasini@unipd.it (G.P.)
* Correspondence: federica.mainente@univr.it

**Abstract:** Twenty-four samples of differently produced commercial Italian cider were analyzed for their polyphenols profile and antioxidant activity. A partial least square regression was used to correlate the ciders' antioxidant activity with their polyphenolic content. Statistical analyses revealed only a clustering pattern in the samples based on their geographical area of production (North-West, NW, vs. North-East, NE). Indeed, NW ciders, compared to NE ones, showed higher antioxidant activity and higher contents of procyanidin B1, catechin, chlorogenic acid, epicatechin, and total polyphenols. On the contrary, no clear-cut clustering pattern determined by cidermaking procedures (i.e., Charmat or Champenoise method) has been observed. These data suggest that the differences observed between NW and NE ciders might be accounted uniquely to the use of different apple varieties and/or the maturation status, as well as the pedoclimatic conditions of their production areas and not for cellar procedures' specificity.

**Keywords:** commercial Italian cider; polyphenols; antioxidant activity; liquid chromatography; geographical clustering

## 1. Introduction

Cider is a popular fermented beverage produced worldwide and made with almost any apple type. The surplus of cooking or dessert apples is used for cidermaking in some regions of the UK, whereas little or no distinction between cider, juice, and dessert apples in Germany is made. On the contrary, French and Spanish cidermakers make a selection of apple varieties named "true cider cultivars" grown for no other purpose except for cider production [1,2]. The fundamental operations of cidermaking are similar, but they may vary among countries [3].

Spain, France, and the United Kingdom are leading European cider producers. The Basque region and Asturias have the most ancient cidermaking tradition in Spain; in France, the main cider cellars are settled in Normandy and Brittany [2]. In Italy, ciders are produced mainly in two distinct areas in the North, i.e., Trentino-Alto Adige and Friuli-Venezia Giulia (North-East), and Piemonte and Valle d'Aosta (North-West) [4]. In southern Italy, limited productions using local apple varieties, such as the Annurca apple, take place [5]. Historical and cultural causes are why cider is not so popular in Italy. Wine production, consumption, and generally wine culture have always dominated in Italy. Moreover, during the fascist period, protectionist laws were promulgated to promote the Italian wine industry and reduce the potential competition from foreign fruit-based fermented beverages, which were heavily taxed [6–8].

Italian ciders are mainly produced with assorted dessert apple varieties' juice [2] following the Charmat (and less frequently, the Champenoise) method, and the final

product is similar to a sparkling white wine (e.g., Prosecco). However, it must be taken into consideration that the organoleptic properties of the beverage are not only influenced by the apple varieties and their characteristics but also by the fermentation conditions and yeast strains used [9] by the production process and the fining treatments [10]. From this point of view, these aspects of Italian cidermaking procedures are scarcely studied in comparison to winemaking ones. As an example, while the autochthonous yeast strains for the valorization of the local typicity (in terms of volatile and chemical profiles) are well studied in wine [11], little information is available on the chemical characteristics of Italian ciders [12–14], and only a few studies have been carried out to explore the effects of apple autochthonous yeast strains [15].

Polyphenols are a heterogeneous group of compounds that affect the organoleptic characteristics of fruit-derived beverages. In particular, they are responsible for color, astringency, and bitterness, and, in general, for the sensory profile of cider. Indeed, they are precursors of volatile compounds, are involved in the fermentative process by controlling the spoilage of bacteria [16], and can also affect technological enzymes (e.g., pectinases) [17]. Polyphenols are also increasingly studied for their beneficial effect on human health. However, the main biological activity of polyphenols is due to their capacity to be antioxidants. This last characteristic can vary due to complex molecular interactions involving synergic and/or antagonistic effects [18,19]. This complexity is increased by the fact that the molecular composition of polyphenols in fruits and vegetables (and their derivatives) can vary depending on the geographical production area [20,21]. Concerning ciders, Alonso-Salces [16] and co-workers were, in fact, able to distinguish about 80% of French ciders assayed according to their region of production by their polyphenol profile.

We have studied 24 Italian ciders by characterizing their polyphenolic composition and measuring their antioxidant activity, and we investigated whether the chemical properties of ciders were related to cidermaking procedures and/or to the specific geographical production origin of the apples. This study could provide the basis for a first chemical characterization of Italian ciders, which to date is still very limited.

## 2. Materials and Methods

Twenty-four Italian ciders commercially available from 11 different brands were considered in this study. The samples were: 1 still cider (no. 1), 17 sparkling ciders produced with the Charmat method (nos. 2–18), and 6 sparkling ciders made following the Champenoise method (nos. 19–24). All the chemical reagents, where not specified, were purchased at Sigma-Aldrich (Steinheim, Germany). The list of reagents is reported in Table S1 of the Supporting Material.

### 2.1. Determination of Total Polyphenolic Contents by the Folin-Ciocalteau Reagent

Total polyphenols were purified from cider samples on a C18 solid-phase cartridge (Waters Corporation, Milford, MA, USA) according to the method described by the manufacturer and quantified by spectrophotometry according to Singleton and Rossi [22]. Total phenols were expressed as mg/L of chlorogenic acid equivalents, as they were the main polyphenols in the samples assayed (see below). Each determination was repeated three times, and results were expressed as mean $\pm$ SD.

### 2.2. Antioxidant Activity

The antioxidant activity (AA) of ciders was measured with two different assays. The possible effect of $SO_2$ on the assays was excluded by treating ciders by sonication and insufflation of $N_2$, whereas the absence of residual ascorbic acid was confirmed by a reverse phase (RP)-HPLC analysis.

The 2,2′-diphenyl-1-picrylhydrazyl (DPPH) assay was performed according to the protocol described by Vakarelova et al. [23] with some modifications. The DPPH solution in methanol (40 mg/L) was prepared daily, and a standard curve was achieved with Trolox (6-hydroxy-2,5,7,8-tetramethylchroman-2-carboxylic acid) with concentrations in the range

of 31.25–1000 μM in methanol. In total, 20 μL of cider or standard was mixed with 200 μL of the DPPH solution, and the decrease in absorbance was measured at 517 nm after 60 min by a microplate reader (Bio-Tek Instruments, Winooski, VT, USA).

The ferric reducing antioxidant power (FRAP) was determined following the protocol described by Picinelli-Lobo et al. [24]. The FRAP reagent was freshly prepared, and a standard curve was obtained with Trolox with concentrations in the range of 31.25–1000 μM in methanol. In total, 30 μL of cider or standard was mixed with 200 μL of the FRAP reagent, and the decrease in absorbance was measured at 595 nm after 40 min by the microplate reader.

The free radical scavenging potential of ciders was determined as described by Dudonnè et al. using 2,2′-azino-bis(3-ethylbenzothiazoline-6-sulfonic acid (ABTS) [25]. The ABTS+● reagent was freshly prepared to an absorbance of $0.7 \pm 0.2$ measured at 734 nm. Trolox with concentrations in the range of 31.25–500 μM in methanol was used as a standard. Then, 20 μL of cider or standard was mixed with 200 μL of the ABTS+● reagent, and the absorbance was measured at 734 nm.

All the experiments were performed in triplicate (n = 3), and the antioxidant activity was expressed in mM of Trolox equivalents.

### 2.3. Other Analyses

Titratable acidity and pH were measured according to the European Union Official Methods of Analysis (Regulation 2676/90) [26]. The alcohol content was claimed on the labels of each sample (Table 1).

**Table 1.** Chemical composition of the Italian ciders assayed.

| Sample | Ethanol % vol | pH | Titratable Acidity [a] | Geographical Origin | Cidermaking Procedure |
|--------|---------------|------|------------------------|---------------------|-----------------------|
| S1 | 4.8 | 3.43 | 3.44 | North-East | Charmat |
| S2 | 7 | 3.3 | 4.89 | North-East | Charmat |
| S3 | 5 | 3.55 | 3.01 | North-East | Charmat |
| S4 | 8 | 3.75 | 3.49 | North-East | Charmat |
| S5 | 8 | 3.79 | 2.88 | North-West | Champenoise |
| S6 | 3 | 3.62 | 4.89 | North-West | Charmat |
| S7 | 4.5 | 3.75 | 3.06 | North-West | Champenoise |
| S8 | 6 | 3.61 | 4.57 | North-West | Charmat |
| S9 | 3.5 | 3.33 | 4.87 | Unknown | Charmat |
| S10 | 8 | 3.48 | 5.42 | North-East | - |
| S11 | 8 | 3.80 | 3.13 | North-West | Champenoise |
| S12 | 8 | 3.18 | 4.92 | North-East | Charmat |
| S13 | 8 | 3.31 | 6.33 | North-West | Champenoise |
| S14 | 8 | 3.76 | 3.11 | North-East | Charmat |
| S15 | 6 | 3.65 | 4.81 | North-East | Charmat |
| S16 | 3.5 | 3.57 | 5.45 | North-East | Charmat |
| S17 | 3 | 3.22 | 3.51 | North-West | Charmat |
| S18 | 5 | 3.14 | 3.80 | North-West | Charmat |
| S19 | 5 | 3.63 | 5.55 | North-East | Charmat |
| S20 | 7 | 3.59 | 5.46 | North-West | Champenoise |
| S21 | 7 | 3.59 | 4.86 | North-East | Charmat |
| S22 | 7 | 3.78 | 3.22 | North-East | Charmat |

**Table 1.** *Cont.*

| Sample | Ethanol % vol | pH | Titratable Acidity [a] | Geographical Origin | Cidermaking Procedure |
|--------|---------------|------|----------------------|---------------------|----------------------|
| S23 | 6 | 3.82 | 4.38 | North-East | Charmat |
| S24 | 8.5 | 3.79 | 3.99 | North-West | Champenoise |
| mean | 6 | 3.56 | 4.29 | | |
| dev.st | 2 | 0.21 | 0.98 | | |
| min | 3.5 | 3.14 | 2.88 | | |
| max | 8.5 | 3.80 | 6.33 | | |

[a] Titratable acidity was expressed in g/L of Malic acid equivalents.

### 2.4. Identification and Quantification of Polyphenols by HPLC

A chromatographic analysis has been performed following the method reported by Lomolino et al. [4]. Briefly, phenolic compounds were separated using an HPLC system (Waters Binary Pump, with 2487 Dual Band Absorbance Detector) equipped with a Kinetex C18 column (4.6 × 150 mm length, 5 μm particle size) from Phenomenex (Torrance, CA, USA). The eluent solvents were water (solvent A) and methanol (solvent B), both containing 0.1% trifluoracetic acid (TFA), and the flow rate was 1 mL/min. The injection volume of each sample was 10 μL. The gradient conditions are indicated in Table S2 (Supporting Material).

Quantification was carried out by the external standard method at 350 nm for chlorogenic acid and the quercetin family compounds and 280 nm for the other phenolics (i.e., tyrosol, epicatechin, catechin, procyanidin B1 and B2, and phloridzin). The integration area measured for each peak was reported in the calibration curve of the corresponding standard. Where no standards were available, molecules belonging to the same family were used for quantification, e.g., quercetin-3-xyloside, quercetin-3-arabinofuranoside, and quercetin-3-arabinopyranoside were quantified as quercetin-3-glucoside equivalents.

### 2.5. Statistical Analysis

A partial least squares (PLS) regression was performed according to the iterative method described by Abdi [27,28]. Cross-validated predictions were obtained using the leave-one-out method. The standard error of regression coefficients and 95% confidence intervals for predictions were estimated by bootstrapping on 10,000 randomly resampled predictors [29]. All calculations were carried out using the software Mathematica (v. 11.2.0.0, Wolfram Research Inc., 100 Trade Center Drive, Champaign, IL, USA).

### 3. Results and Discussion

The chemical characteristics of the analyzed ciders are shown in Table 1, where a certain degree of variability can be appreciated among the measured parameters (see, e.g., titrable acidity and ethanol content). This variability is similar to that reported in the literature for ciders produced abroad [30–34], and it is a consequence of the apple varieties, the climatic and seasoning factors, and the technical procedures of cidermaking [35,36]. All these variables should contribute to the typical features of the product affecting the sensory characteristics of the beverage alone or in combination with other molecules, in particular, if food is assumed contemporaneously [4]. For example, Alonso-Salces [16] described that the French and Basque ciders' organoleptic properties differ markedly due to their production technologies. Italian ciders are produced following the white sparkling winemaking protocol [37], where the pressing steps take place just after the grinding. Furthermore, sulfur dioxide and other preservatives (e.g., lysozyme as described by Mainente et al. [38]) are added to Italian commercial ciders to avoid spoilage bacteria growth and malolactic fermentation (desirable in French and Spanish ciders) [37,39,40].

### 3.1. Polyphenols Profiles of the Commercial Italian Ciders

The RP-HPLC analysis carried out for all the cider samples identified 14 polyphenols belonging to the group of phenolic acids (chlorogenic acid), flavan-3-ols (catechin, epicatechin, procyanidins B1 and B2), flavonols (quercetin, quercetin-3-glucoside, quercetin-3-rhamnoside, quercetin-3-xyloside, quercetin-3-galactoside, quercetin-3-arabinofuranoside, and quercetin-3-arabinopyranoside), dihydrochalcone (phloridzin), and the group of volatile phenols (tyrosol). The concentration of phenolic compounds is reported in Table 2.

**Table 2.** Concentration (mg/L) of polyphenols in analyzed Italian ciders.

| | **Min** | **Max** | **Mean** | **Dev.st** |
|---|---|---|---|---|
| Phenolic acids | | | | |
| Clorogenic acid (350 nm) | 0.93 | 188.04 | 37.62 | 45.81 |
| Flavan-3-ols | | | | |
| Procyanidin B1 | 0.49 | 11.34 | 2.76 | 2.47 |
| Catechin | 1.59 | 16.19 | 7.14 | 4.36 |
| Procyanidin B2 | 0.03 | 29.15 | 11.54 | 10.51 |
| Epicatechin | 0.02 | 58.16 | 21.88 | 19.79 |
| Dihydrochalcone | | | | |
| Phloridzin (280 nm) | 0 | 39.17 | 12.76 | 10.83 |
| Flavonols | | | | |
| Quercetin-3-arabinofuranoside | 0 | 4.33 | 0.78 | 1.05 |
| Quercetin-3-xyloside | 0 | 2.70 | 0.86 | 0.64 |
| Quercetin-3-arabinopiranoside | 0 | 4.77 | 0.97 | 1.04 |
| Quercetin-3-glucoside (350 nm) | 0 | 4.90 | 1.05 | 1.02 |
| Quercetin | 0.61 | 4.57 | 1.73 | 0.91 |
| Quercetin-3-rhamnoside (350 nm) | 0 | 12.70 | 3.30 | 2.46 |
| Hyperoside (quercetin-3-galactoside) | 0 | 30.52 | 8.90 | 7.47 |
| Volatile phenols | | | | |
| Tyrosol | 0 | 14.16 | 5.51 | 3.40 |

The integration area measured for each peak was reported in the calibration curve of the corresponding standard.

In general, the concentration of each phenol detected revealed wide ranges, as indicated by their standard deviations. The most abundant family in the Italian ciders is the flavan-3-ols group, which is mainly represented by catechin (16.5%), epicatechin (50.5%), and procyanidins B1 (6.4%) and B2 (26.6%). The average concentration of the flavan-3-ols detected in the Italian ciders is about 43 mg/L. The amount of these compounds is reported to be slightly lower in the Asturian ciders [41], whereas the Basque and French ciders show higher levels of flavan-3-ols up to 389 mg/L [16]. Alonso-Salces et al. [16] explained the differences in the polyphenolic profiles in relation to the cidermaking procedures used for their production. However, even if the technologies used in France and Italy are similar, the concentration of total flavan-3-ols in the Italian beverage is nine times lower than that reported for the French ones [16]. Thus, we hypothesize that this difference could be mainly related to the apple cultivars. Indeed, French cidermakers usually make a selection of "true cider cultivars" that are grown with the sole purpose of producing cider, whereas in Italy, ciders are mainly produced with a blend of dessert apples that are described to have a lower level of phenolic compounds in comparison to the ones for cider [42].

The chlorogenic acid is the only polyphenol belonging to the phenolic acid group that has been detected in the Italian ciders with a mean concentration of 37.6 mg/L. This concentration is different from that reported in the literature for other ciders [16,41]. The

phenolic acid family is the most important group of polyphenols in the Asturian ciders, representing around 60%. However, the amount of chlorogenic acid in other Spanish ciders is meager, probably as a consequence of oxidation and hydrolysis with reduction into hydroxycinnamic acids' derivatives during fermentation and post-fermentation [43]. Although Spanish regulations allow the use of sulfur dioxide, fermentation is usually carried out without its addition [32]. On the contrary, as described above, the Italian ciders are produced following the winemaking protocol, which takes into account the addition of $SO_2$ in the early stage of apple crushing and pressing, and this might inhibit the oxidation of polyphenols. The high content of chlorogenic acid and the absence of the corresponding derivatives caffeic/hydrocaffeic and p-coumaric/hydrocoumaric acids could also be explained by the use of $SO_2$ [44]. However, Fratianni et al. [5] described a cider produced in southern Italy with high levels of polyphenols belonging to the hydroxycinnamic acid group (i.e., caffeic, ferulic, and p-coumaric acids). This might be ascribed to the long fermentation (up to 3 weeks) of the crushed apples used to produce the beverage. As reported before, the use of $SO_2$ avoids the natural fermentation by wild yeasts, while spontaneous fermentation is still performed in many countries such as Spain, France, and Ireland [45,46]. It must be noted that these yeasts constitute a complex population that includes species different from those responsible for wine-must fermentation [46,47]. The microbiota of apple juice could have different enzymatic pools, influencing the final cider both in terms of specific aroma and beverage spoilage [48–50].

Phloridzin and phloretin are described as the most relevant bioactive polyphenols in apples, and they have been found exclusively in apples and their derivatives so that they could be helpful for food authenticity studies [51]. Thirteen out of the twenty-four ciders assayed showed a concentration of phloridzin higher than 10 mg/L. This value agrees with those quantified for Asturian, Basque, and French ciders [16,41]. On the contrary, phloretin, the aglycone of phloridzin, has been detected in small quantities (1–5 mg/L) in ciders produced abroad [16,41,52], whereas in Italian cider, it was virtually absent. This could depend on the breaking of glycosidic bonds during bacterial metabolism. Indeed, Laaksonen et al. [52] described that the concentration of phloretin increases as long as fermentation proceeds. In support of this hypothesis, it is worth noting that malolactic fermentation is usually avoided in Italian ciders [37].

Italian ciders showed an average concentration of 17.6 mg/L of total flavonols, which is up to three times higher than that reported for the French and Spanish ciders [16,41]. The glycosylated compounds of the group account for 90% of total flavonols, and quercetin-3-rhamnoside and hyperoside account for about 77% of glycosylated quercetin (Table 3). This ratio is similar to that reported for the Asturian ciders [41]. The high amount of glycosylated compounds could be related to the faster fermentation of Italian ciders in comparison to those produced abroad.

Tyrosol is a volatile phenol formed starting from tyrosine by yeast metabolism, and its concentration is related to the length of fermentation. Tyrosol was not revealed in two out of the twenty-four ciders assayed, but values ranging from 2.9 to 14.2 mg/L have been detected in the remaining samples. The average value of the tyrosol concentration (5.5 mg/L) is lower than that reported for Asturian and Basque ciders and other fermented beverages, such as wine and beer [41,53]. This feature might be explained by the fact that Italian ciders are, in general, characterized by a few days of fermentation and by a bentonite treatment of apple juice to eliminate proteins. This practice, aimed at increasing the stability of cider [37], has the effect of reducing protein availability for yeast metabolism. Furthermore, this could explain the lack of differences in tyrosol content between ciders produced by the Charmat or the Champenoise methods.

The polyphenolic profiles of Italian ciders showed differences when compared to data reported for ciders produced in other countries [16,41,52,54]. This result is not surprising because the concentration of polyphenols varies according to the raw materials (i.e., apple varieties, fruit maturity, and cultivation condition) and the cidermaking procedures, as described above.

**Table 3.** Antioxidant activity of the Italian ciders assayed.

| Sample | FRAP [a] | DPPH [a] | ABTS [a] | PFT [b] |
|---|---|---|---|---|
| S1 | 8.58 | 8.75 | 2.66 | 149.05 |
| S2 | 9.02 | 9.13 | 2.59 | 141.35 |
| S3 | 7.86 | 8.13 | 2.65 | 150.67 |
| S4 | 8.88 | 9.01 | 2.87 | 136.36 |
| S5 | 14.19 | 13.56 | 5.10 | 377.16 |
| S6 | 11.83 | 11.54 | 3.10 | 338.69 |
| S7 | 21.08 | 19.47 | 4.62 | 743.07 |
| S8 | 10.74 | 10.61 | 3.47 | 254.87 |
| S9 | 13.16 | 12.68 | 5.51 | 475.42 |
| S10 | 8.63 | 8.8 | 2.99 | 144.73 |
| S11 | 13.52 | 12.99 | 5.18 | 387.01 |
| S12 | 9.86 | 9.85 | 3.32 | 169.16 |
| S13 | 8.16 | 8.40 | 1.85 | 126.78 |
| S14 | 10.66 | 10.54 | 2.67 | 194.67 |
| S15 | 12.47 | 12.09 | 3.47 | 268.64 |
| S16 | 14.61 | 13.92 | 3.93 | 344.36 |
| S17 | 10.36 | 10.28 | 2.78 | 294.82 |
| S18 | 16.08 | 15.18 | 4.19 | 602.29 |
| S19 | 7.58 | 7.90 | 2.69 | 154.18 |
| S20 | 9.94 | 9.92 | 3.89 | 284.29 |
| S21 | 7.47 | 7.80 | 1.94 | 93.17 |
| S22 | 7.94 | 8.20 | 4.86 | 104.37 |
| S23 | 9.33 | 9.40 | 2.73 | 221.39 |
| S24 | 10.86 | 10.70 | 4.68 | 236.51 |
| mean | 10.95 | 10.79 | 3.49 | 266.38 |
| dev.st | 3.24 | 2.78 | 1.10 | 161.98 |
| min | 7.47 | 7.80 | 1.88 | 93.17 |
| max | 21.08 | 19.47 | 5.51 | 743.07 |

[a] Antioxidant activity was expressed in mM of Trolox equivalents; [b] total polyphenol content was expressed in mg/L of chlorogenic acid equivalents.

### 3.2. Antioxidant Activity of Ciders

Phenolics, vitamins, and carotenoids are the three major groups of compounds that contribute to the antioxidant activity of foods [55]. Regarding cider, polyphenols are the molecules mainly involved in the antioxidant activity (AA) of the beverage. The AA is frequently estimated by several methods, including DPPH, FRAP, and ABTS, because it has been reported that these protocols have shown different results across laboratories and among samples [55,56].

The assays need the optimization of the reaction time, as it is dependent on the kind of sample analyzed. The steady state of DPPH and FRAP analyses was reached at 60 min and 40 min, respectively. The absorbances of ABTS assay were read after 10 min. Table 3 shows the AA values obtained by the assays. Both the DPPH and FRAP assays showed values ranging from about 7 mM to 21 mM of Trolox equivalents, revealing superimposable values (Table 3). For this reason, the DPPH data were excluded from the following analyses.

Values determined by the ABTS protocol showed an AA ranging from about 2 mM to 5.5 mM of Trolox equivalents (Table 3).

### 3.3. Statistical Analysis

A partial least square (PLS) regression was used to test whether the antioxidant activity of the cider samples could be explained by their polyphenolic content [27]. We considered as the independent variables, the concentrations of 15 polyphenols (including the total polyphenol content) measured in the 24 ciders assayed, and as the dependent variables, the antioxidant activity of each cider as measured by two methods, namely FRAP and ABTS. As described above, the results obtained with the DPPH protocol have not been considered, as they were strongly superimposable with those obtained with the FRAP method. Figure 1A shows the root mean squared error (i.e., the mean distance between observed and predicted values, see Kiralj and Ferreira [57] for details) for the whole dataset (RMSE), and cross-validation (RMSECV) as the function of increasing latent variables. Optimal predictions were achieved when the model took into account seven latent variables. Accordingly, the cross-validated correlation coefficient [57] was maximal for the same number of latent variables (Figure 1B). The cumulative percentage of explained variance in both the independent and dependent variables by latent variables is shown in Figure 1C.

This preliminary analysis suggested considering a model with the first seven latent variables as optimal for the PLS regression. Indeed, the model provided good predictions of the antioxidant activity of ciders (see Figure 2). The PLS regression coefficients are given in Table 4. Some polyphenols (i.e., chlorogenic acid, epicatechin, and phloridzin) have a negative coefficient, thus suggesting that these molecules might have a pro-oxidant effect. However, this coefficient also takes into account all the other components. Therefore, a negative coefficient does not necessarily indicate an antagonistic effect on the antioxidant power, but it could hide a relationship with other components. Additionally, some polyphenols could be precursors of molecules that have more antioxidants, so the higher amount of a precursor could negatively correlate with the antioxidant power without being a pro-oxidant. Moreover, it is known that, under specific conditions and concentrations, some polyphenols might act as pro-oxidants [58–60] or they show additive, synergic, and/or antagonistic effects [60].

One striking feature of the PLS regression is that it allows an exploratory analysis of data sets. Figure 3 combines different features of the PLS regression. It shows a map where the correlation between the Y variables and the latent variables for the first two dimensions is superimposed to the correlations for the X predictors and the observations (ciders) projected in the same space of the first two latent variables.

**Table 4.** PLS regression parameters for the model with the selected latent variables.

| Var. # | Predictors | [a] FRAP | [a] ABTS |
|---|---|---|---|
| | [b] Intercept | $0.113 \pm 1.0{\cdot}10^{-3}$ | $0.041 \pm 4.3{\cdot}10^{-4}$ |
| 1 | Procyanidin B1 | $0.137 \pm 1.5{\cdot}10^{-3}$ | $0.003 \pm 7.9{\cdot}10^{-4}$ |
| 2 | Catechin | $0.277 \pm 2.5{\cdot}10^{-3}$ | $0.048 \pm 1.6{\cdot}10^{-3}$ |
| 3 | Procyanidin B2 | $0.162 \pm 7.9{\cdot}10^{-4}$ | $0.042 \pm 4.9{\cdot}10^{-4}$ |
| 4 | Clorogenic acid | $-0.039 \pm 4.1{\cdot}10^{-4}$ | $-0.013 \pm 2.3{\cdot}10^{-4}$ |
| 5 | Epicatechin | $-0.043 \pm 5.4{\cdot}10^{-4}$ | $0.017 \pm 3.5{\cdot}10^{-4}$ |
| 6 | Phloridzin | $-0.204 \pm 1.7{\cdot}10^{-3}$ | $-0.053 \pm 1.0{\cdot}10^{-3}$ |
| 7 | Quercitin-3-glucoside | $0.080 \pm 8.1{\cdot}10^{-4}$ | $0.023 \pm 4.0{\cdot}10^{-4}$ |
| 8 | Quercitin-3-rhamnoside | $0.416 \pm 2.4{\cdot}10^{-3}$ | $0.110 \pm 1.6{\cdot}10^{-3}$ |
| 9 | Quercitin-3-xyloside | $0.072 \pm 4.0{\cdot}10^{-4}$ | $0.026 \pm 2.2{\cdot}10^{-4}$ |

**Table 4.** *Cont.*

| Var. # | Predictors | [a] FRAP | [a] ABTS |
|---|---|---|---|
| 10 | Quercitin-3-arabinofuranoside | $0.005 \pm 1.5 \cdot 10^{-3}$ | $0.038 \pm 7.8 \cdot 10^{-4}$ |
| 11 | Quercitin-3-arabinopiranoside | $0.060 \pm 7.3 \cdot 10^{-4}$ | $0.011 \pm 4.2 \cdot 10^{-4}$ |
| 12 | Hyperoside | $0.290 \pm 2.2 \cdot 10^{-3}$ | $0.096 \pm 1.2 \cdot 10^{-3}$ |
| 13 | Tyrosol | $0.039 \pm 1.5 \cdot 10^{-3}$ | $0.069 \pm 9.3 \cdot 10^{-4}$ |
| 14 | Quercitin | $0.175 \pm 2.8 \cdot 10^{-3}$ | $0.058 \pm 1.1 \cdot 10^{-3}$ |
| 15 | PFT | $0.022 \pm 1.3 \cdot 10^{-4}$ | $0.004 \pm 6.4 \cdot 10^{-5}$ |

[a] Median values $\pm$ SE as calculated by bootstrapping on 10,000 random samples; [b] the coefficients are expressed in original data units, i.e., mM Trolox equivalents (intercept) and mg/L (all the coefficients except the intercept).

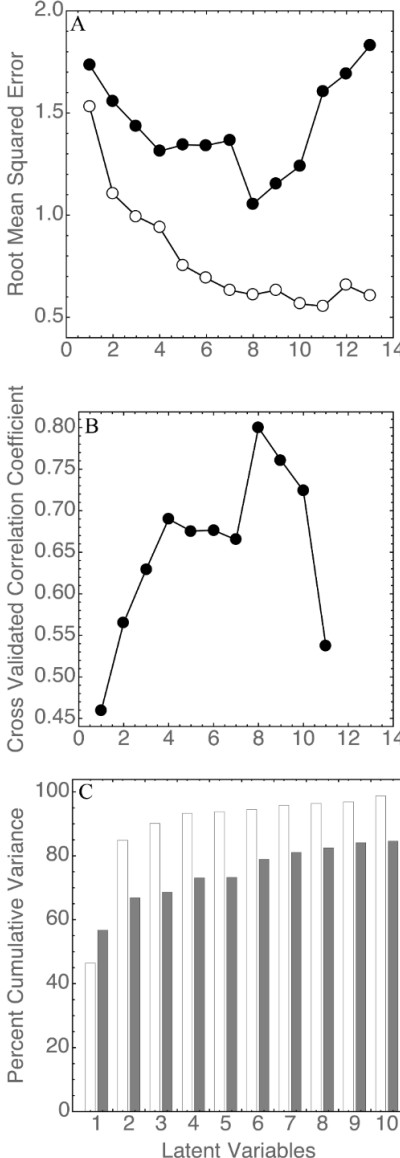

**Figure 1.** Selection of the number of latent variables for PLS regression. Panel (**A**): root mean squared error of PLS regression carried out on original data (white circles) and of cross-validation (black circles) as the function of the number of latent variables. Panel (**B**): cross-validated correlation coefficient. Panel (**C**): percent cumulative explained variance in X (the matrix of predictors, white bars) and in Y (the matrix of dependent variables, grey bars) using the first seven latent variables.

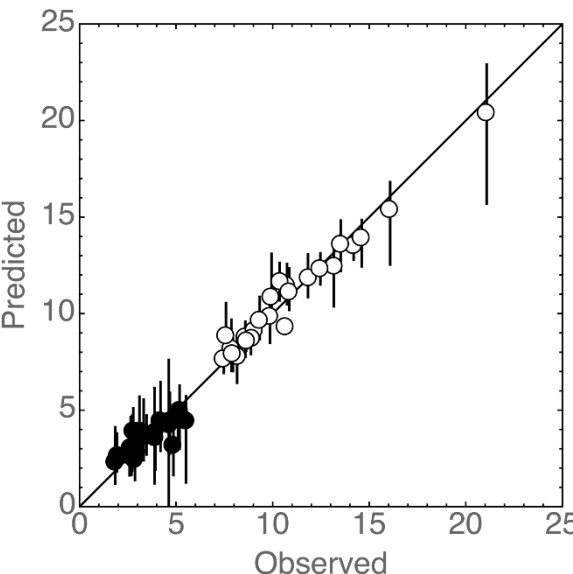

**Figure 2.** Predicted vs. observed antioxidant activity. Bars represent 95% confidence intervals. White circles: antioxidant activity as measured by FRAP; black circles: antioxidant activity as measured by ABTS.

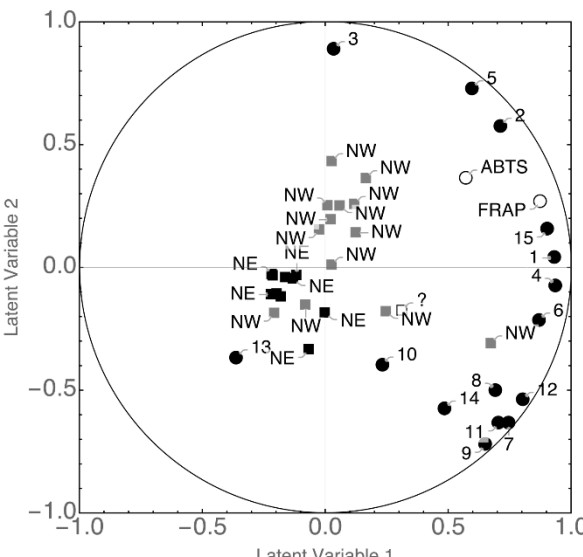

**Figure 3.** Map of the correlations between predictors (black circles) and dependent variables (white circles) with the first two latent vectors and projections of the observations onto the same space. The predictors have been coded with numbers ranging from 1 to 15 as defined in Table 4. Ciders have been coded depending on the geographical area of their production (NE, black square; NW, grey square). The place of production of one cider sample is unknown (white square).

The ciders were coded depending on the geographical area of their production (North-East or North-West, see Table 2), and we compared them for their polyphenols content and antioxidant activity, and the results were statistically significant as evaluated by the Mann–Whitney test (Figure 4). The correlation map in Figure 3 shows an apparent clustering pattern, with NE ciders clustering together in the opposite direction of the vectors representing antioxidant activity and those of various polyphenols. Indeed, NW ciders showed higher antioxidant activity and higher contents of procyanidin B1, catechin, chlorogenic acid, epicatechin, and total polyphenols content as compared to NE ciders (Figure 4).

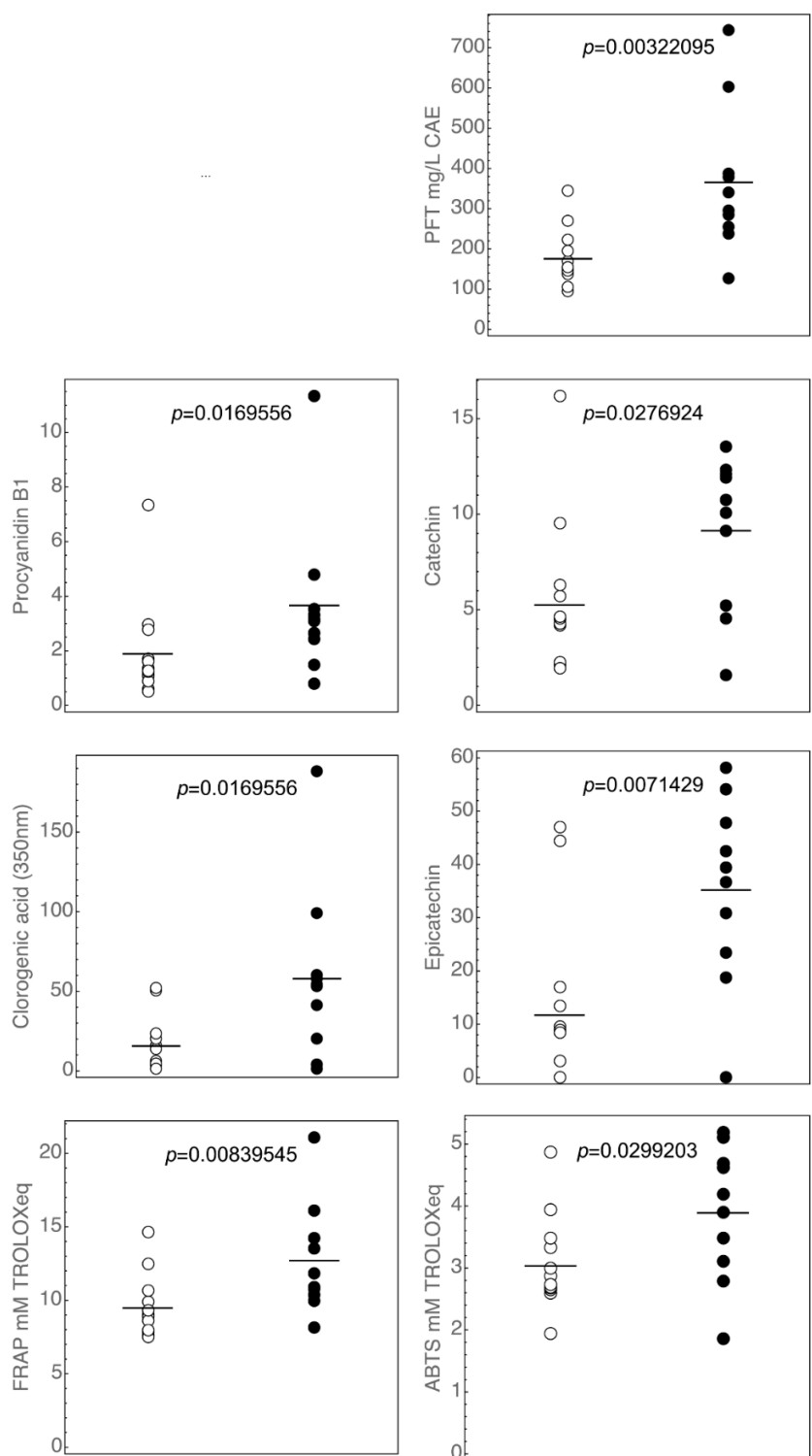

**Figure 4.** Differences between NE (white circles) and NW (black circles) ciders. Only statistically significant differences are shown. *p* values reported in each plot have been calculated using the Mann–Whitney test.

Ciders produced in neighboring geographical areas may show chemical differences [16,32] that have been related to the different traditional cidermaking protocols used. However, we could not observe any clear-cut clustering pattern in our samples on the basis of cidermaking procedures (i.e., Charmat or Champenoise method, data not shown). Thus, the geographical clustering of Italian ciders might be the result uniquely of the apple used

(e.g., varieties and maturation) and the pedoclimatic conditions of the area, while the potential distinctiveness of local Italian cidermaking is not exploited. Nicolini G. et al. [61], for example, described the possibility to obtain different products with well-characterized compositional profiles by using six different dessert apple varieties (Golden Lasa, Braeburn, Granny Smith, Fuji, and Reinette de Champagne, grown in a single plot in the Adige Valley, and Reinette du Canada from the Non Valley) cultivated in the Trentino Alto Adige region (North-Eastern Italy).

Lorenzini M. et al. [15], studying the autochthonous yeast microbiota of apple juice, evidenced deep differences in terms of species composition between two geographical cider production areas. These authors underlined that improving knowledge of yeast diversity in the cider cellar, which has received little attention when compared to wine production, can contribute to improving the typicity of the products.

## 4. Conclusions

The analyses performed on 24 commercial Italian ciders showed that their antioxidant activity is higher than that observed for cider produced in other countries, whereas the total polyphenol content of the ciders assayed is on average lower. This might appear contradictory. We note the following: 1. There is no general consensus on the use of a given molecular standard for quantification purposes. Different studies indeed report data that refer to different standards [24,30,54]. 2. Polyphenols may interact, showing either synergic or antagonistic effects depending on the experimental conditions and concentrations [60], and thus they may contribute differently to the overall antioxidant potential of a solution. Thus, a direct comparison of the data obtained in different studies may have puzzling results.

Our main goal has been to look for possible correlations between the polyphenols profile of Italian ciders; their antioxidant potential; and various chemical-physical, methodological, and geographical parameters. Differently to other reports, we found no correlations between ciders' antioxidant activity and cidermaking procedures, i.e., Charmat or Champenoise production methods. However, we did find a clear dependence of antioxidant activity on the geographical production areas. We, therefore, conclude that the observed variability of the analyzed ciders is likely to depend only upon the different varieties of apple used, their maturation levels, and the different pedoclimatic conditions of the production areas. How much these factors contribute to the antioxidant potential of the ciders will have to be investigated. In any case, the autochthonous apple and yeast bio-diversity and local cidermaking Italian area should be exploited better to obtain more typified products. Traditional Italian agri-food products are usually characterized by extreme compositional and organoleptic variability. Examples are cheeses, cured meats, and wines. This result is the consequence of attachment to traditions, which is typical of the Italian population and the historical fragmentation of the territory into numerous sovereign states connected only in the recent past. These two factors led to developing different production strategies typical of every town. The policy of wine safe-guarding, as the national beverage, has probably led to the loss of the traditions of preparing other fermented drinks, such as cider.

**Supplementary Materials:** The following supporting information can be downloaded at: https://www.mdpi.com/article/10.3390/beverages9020054/s1, Table S1: List of the chemical reagent used. Table S2: Gradient conditions for polyphenols separation by HPLC.

**Author Contributions:** Conceptualization, F.M., C.R., S.V. and G.P.; formal analysis, F.M. and S.V.; resources, G.P. and S.V.; data curation, F.M. and C.R.; writing—original draft preparation, F.M. and C.R.; writing—review and editing, F.M., C.R., S.V. and G.P.; funding acquisition, G.P. and S.V. All authors have read and agreed to the published version of the manuscript.

**Funding:** This work was supported by the University of Padova (Grant Number CPDA148173).

**Data Availability Statement:** Not applicable.

**Acknowledgments:** We are grateful to Roberto Chignola (Department of Biotechnology, University of Verona, Strada Le Grazie 15, 37134 Verona, Italy) for his help in conducting the statistical analysis. All individuals included in this section have consented to the acknowledgement.

**Conflicts of Interest:** The authors declared that they have no conflict of interest.

**Abbreviations**

NE: North-East; NW, North-West; AA, antioxidant activity; DPPH, 2,2′-diphenyl-1-picrylhydrazyl; Trolox, 6-hydroxy-2,5,7,8-tetramethylchroman-2-carboxylic acid; FRAP, ferric reducing antioxidant power; ABTS, 2,2′-azino-bis(3-ethylbenzothiazoline-6-sulphonic acid); TFA, trifluoroacetic acid; PLS, partial least squares; RMSE, root mean square error; RMSECV, root mean square error cross-validation.

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
