# Peer review of "Polyphenols Profile and Antioxidant Activity Characterization of Commercial North Italian Ciders in Relation to Their Geographical Area of Production and Cidermaking Procedures"

_beverages, doi:10.3390/beverages9020054_

Round 1

Reviewer 1 Report

I read and carefully reviewed the manuscript. In my opinion, the authors did a good job at explaining the methods and discussing the results of the experiments. The conclusions are consistent with the results and properly compared to previous reports. The topic is interesting and relevant. I would recommend authors review values in tables an edit them (if necessary) so they are consistent in terms of the significant figures reported. 

The manuscript is clear and easy to follow, i didn't find any major issues with the english grammar. 

Author Response

Dear Editor,

We decided to apply a revision to our paper Ref. No beverages-2348137 entitled: "Polyphenols Profile and Antioxidant Activity Characterization of Commercial North Italian Ciders in Relation to Their Geographical Area of Production and Cidermaking Procedures” following the Reviewers and Editor’s suggestions, to improve the manuscript quality.

We have carefully considered all the raised questions and suggestions. The modifications in the revised form are marked with the track changes function.

Answer to reviewers

We thank the Editor and the Reviewers and for their comments that allowed us to improve the quality of our manuscript.

Answer to Reviewer 1:

Reviewer #1: I read and carefully reviewed the manuscript. In my opinion, the authors did a good job at explaining the methods and discussing the results of the experiments. The conclusions are consistent with the results and properly compared to previous reports. The topic is interesting and relevant. I would recommend authors review values in tables and edit them (if necessary) so they are consistent in terms of the significant figures reported. The manuscript is clear and easy to follow, I didn't find any major issues with the english grammar

Authors: we thank the Reviewer for his/her comment and suggestion. We provided to review all the tables.

Reviewer 2 Report

1. HPLC chromatograms for both standard of the 14 polyphenolic compounds and samples of cider samples.

2. Do the standards for the 14 phenolic compounds are available?

2. more discussion of the results is needed

Author Response

Dear Editor,

We decided to apply a revision to our paper Ref. No beverages-2348137 entitled: "Polyphenols Profile and Antioxidant Activity Characterization of Commercial North Italian Ciders in Relation to Their Geographical Area of Production and Cidermaking Procedures” following the Reviewers and Editor’s suggestions, to improve the manuscript quality.

We have carefully considered all the raised questions and suggestions. The modifications in the revised form are marked with the track changes function.

Answer to reviewers

We thank the Editor and the Reviewers and for their comments that allowed us to improve the quality of our manuscript.

Answer to Reviewer 2:

Reviewer  #2: Comments and Suggestions for Authors

  1. HPLC chromatograms for both standard of the 14 polyphenolic compounds and samples of cider samples.

Authors: We can provide, if the reviewer want it, the chromatograms of the standards (to be added in Supplemetary material), but we think adding the chromatograms of all the 24 cider samples would be too dispersive

  1. Do the standards for the 14 phenolic compounds are available?

Authors: The commercial standards are available for almost all the identified compounds, except for the quercetin derivatives 3-xyloside, 3-arabinofuranoside and 3-arabinopiranoside that were identified based on their retention time and spectra characteristics by comparison with literature data, as reported in reference 4 (Lomolino et al. 2022).

  1. more discussion of the results is needed

Authors: we slightly modified the paper throughout the text.

Reviewer 3 Report

From my point of view, the article intitled " Polyphenols Profile and Antioxidant Activity Characterization of Commercial North Italian Ciders in Relation to Their Geographical Area of Production and Cidermaking Procedures " presents numerous drawbacks that need to be fixed.

In the manuscript is not well identify their interest or their highlights.

In my opinion the study is not well organized and miss some importations, explanation and better support/discuss the statistical analysis.

The samples collected are representatives?? What is the interest to study the Cider Geographical Area of Production, if is not possible to know the Geographical Area of raw material use in their production, varieties of apples or maturity stage?

The design is not quite correct or is unwell explained.

What is the importance of the cider polyphenols content or antioxidant properties separated from other important characteristics of the cider? Namely sensory characteristics, between others.

If the concentration of polyphenols varies according to raw materials (i.e., apple varieties, fruit maturity, and cultivation condition) and the cidermaking procedures, how can the author conclude something if the study are not delineated according these objectives?

There are several repetitions in the manuscript, and some inaccuracies in the statistical analysis

The author refers that “The analyses performed on 24 commercial Italian ciders showed that their antioxidant activity is higher than that observed for cider produced in other countries, whereas the total polyphenol content of the ciders assayed is on average lower.” There are a lot of explanation for that, even the methods used in this study, and not all polyphenols have antioxidant activity.

How the author confirmed if the peaks are well identified? They used liquid mass analysis?? Are you sure that no more than one compound has the same retention time?

More comments in the attached file

Author Response

Dear Editor,

We decided to apply a revision to our paper Ref. No beverages-2348137 entitled: "Polyphenols Profile and Antioxidant Activity Characterization of Commercial North Italian Ciders in Relation to Their Geographical Area of Production and Cidermaking Procedures” following the Reviewers and Editor’s suggestions, to improve the manuscript quality.

We have carefully considered all the raised questions and suggestions. The modifications in the revised form are marked with the track changes function.

Answer to reviewers

We thank the Editor and the Reviewers and for their comments that allowed us to improve the quality of our manuscript.

Answer to Reviewer 3:

Reviewer  #3: From my point of view, the article intitled " Polyphenols Profile and Antioxidant Activity Characterization of Commercial North Italian Ciders in Relation to Their Geographical Area of Production and Cidermaking Procedures " presents numerous drawbacks that need to be fixed.

In the manuscript is not well identify their interest or their highlights.

Authors: A sentence explaining the importance of the study has been added (lines 73-74).

In my opinion the study is not well organized and miss some importations, explanation and better support/discuss the statistical analysis.

Authors: The manuscript has been improved.

The samples collected are representatives?? What is the interest to study the Cider Geographical Area of Production, if is not possible to know the Geographical Area of raw material use in their production, varieties of apples or maturity stage?

Authors: In Italy, the more important producing areas are North-East and North-West, and the collected samples are almost equally distributed in the two different regions (13 vs 11 samples, respectively). Regarding the distribution of samples based on the cidermaking procedure, it reflects the real distribution of the product on the market in Italy, where the main procedure to obtain cider is the Charmat method. Concerning the interest of the study related to the Geographical Area, a sentence has been added at the end of the paper (lines 385-392).

The design is not quite correct or is unwell explained. What is the importance of the cider polyphenols content or antioxidant properties separated from other important characteristics of the cider? Namely sensory characteristics, between others.

Authors: The referee is right in fact we stated in lines 55-58 that in cider the polyphenols are considered responsible of the main organoleptic characteristics. However, it was considered not informative to perform also sensory analysis on our samples, due to the very different content of residual sugars, which would have masked most of the other sensory characteristics. We didn’t measure the sugar content, but we know that in sparkling wine production with Charmat method (which is the most represented among our samples) it is possible to stop the fermentation in different moments, obtaining different content of unfermented sugars. This can be confirmed by the average alcohol content, which is lower in samples obtained with Charmat method compared to those obtained with Champenoise method.

If the concentration of polyphenols varies according to raw materials (i.e., apple varieties, fruit maturity, and cultivation condition) and the cidermaking procedures, how can the author conclude something if the study are not delineated according these objectives?

Authors: Again refer to the sentence added at lines 385-392.

There are several repetitions in the manuscript, and some inaccuracies in the statistical analysis

Authors: the repetition have been eliminated as well the inaccuracies in the statistical analysis.

The author refers that “The analyses performed on 24 commercial Italian ciders showed that their antioxidant activity is higher than that observed for cider produced in other countries, whereas the total polyphenol content of the ciders assayed is on average lower.” There are a lot of explanation for that, even the methods used in this study, and not all polyphenols have antioxidant activity.

Authors: The methods for the determination of antioxidant activity are all data expressed as Trolox equivalents, which guarantee the possibility to compare our results with those of other studies. One possibility of such data could be the interference of different SO2 contents (which is an antioxidant as well) in the samples, but this interference was carefully avoided (use of C18 cartridges, lines 85-87, and sonication+gas sparging, lines 93-95). A possible explanation is the interaction of different polyphenol compounds, that, as reported in lines 309-312, can show additive, synergic or antagonistic effect among them.

How the author confirmed if the peaks are well identified? They used liquid mass analysis?? Are you sure that no more than one compound has the same retention time?

Authors: We didn’t use a liquid mass; however the peaks were identified (see reference 4, of the same research group) with a Diode Array detector, which allowed to identify the purity of the molecules not only by their retention time but also by their spectrum. The overlapping of different molecules would have been detected by a change in the spectrum.

More comments in the attached file

We thanks the reviewer for his/her indications. All the correction have been done.

Line 34: the reference has been adjusted

line 36: a reference has been added

line 43: the reference has been adjusted

line 71: “similarly to wines” has been removed

lines 70-73: the sentence has been modified

line 74: the Materials and Methods section has been modified and the information required about reagents (cas number, purity and company) have been reported in Table 1S of the Supporting Material.

line 91: RP-HPLC has been modified

lines 92-98: we don’t understand the request of the referee. Protocols are reported for all the techniques, and it is also reported that the data were expressed in Trolox equivalents.

lines 99-104:  the sentence has been improved 

lines 105-109: the sentence has been improved

line 112: all the products were commercially available, then, for law, the label should report the accurate alcohol content of the product, which should be obtained by official and controlled methods. Even in case of a small error due to the rounding to the 0.5%, the alcohol content was considered just as an additional information, but was not included in the statistical analysis.

Line 121: Table 1 has been moved to the Supporting Material file. Table 1 is now Table 2S. 

Line 122: a part of the sentence has been removed

Line 153: Table 2 has been modified. Table 2 is now Table 1.

Line 154: a part of the sentence has been removed

Line 165: Table 3 and its legend has been modified. 

Line 166: the sentence has been removed

Lines 171-173:  the sentence has been removed

Line 176: reference number has been added

Line 187: the sentence has been removed

Line 212: the sentence has been removed

Line 250: the review is right. The paragraph has been modified and a table with the antioxidant activity of the ciders analysed has been added

Line 282: This type of analysis allows to fix the number of latent variables necessary to model the data on the basis of the known statistical considerations mentioned in section 3.3 (Kiralj et al. 2009). The legend of Figure 1 has been moved.

Line 289: Parameters of the PLS regression model are shown in Tab. 4 along with the error estimated by bootstrapping on 10000 random samples.

Line 302: As mentioned in the manuscript (section 3.2) the AA obtained with DPPH assay was not considered in the PLS regression model since values were superimposable to the ones obtained with FRAP assay. The assays used (i.e. ABTS and FRAP) are carried out at acidic (FRAP) and neutral (ABTS) conditions. So the pH values may have an important effect on the reducing capacity of antioxidants. Due to the complexity of the composition of ciders, separating each antioxidant compound and studying it individually it would be costly and inefficient, we decided to evaluate the AA of our sample taking into account different protocols. The PLS model was developed to explain simultaneously the two variables (i.e. FRAP and ABTS) as a function of the observation.

Line 305: Table 4 and its legend has been modified.

Line 314: Figure 1C shows the cumulative percentage of variance in the data, explained by the latent variables and from which the percentage explained by each variable can be inferred. The text has been modified

Line 324: the adjective has been changed.
